# The Intransitive Logic of Directed Cycles and Flipons Enhances the Evolution of Molecular Computers by Augmenting the Kolmogorov Complexity of Genomes

**DOI:** 10.3390/ijms242216482

**Published:** 2023-11-18

**Authors:** Alan Herbert

**Affiliations:** InsideOutBio, 42 8th Street, Charlestown, MA 02129, USA; alan.herbert@insideoutbio.com

**Keywords:** evolution, flipons, kolmogorov complexity, dissipative structures, hypercycles, directed cycles, intransitive logic, peptide patches, junk DNA, DNA repeats, microRNA, Alu, condensate, free energy, entropy

## Abstract

Cell responses are usually viewed as transitive events with fixed inputs and outputs that are regulated by feedback loops. In contrast, directed cycles (DCs) have all nodes connected, and the flow is in a single direction. Consequently, DCs can regenerate themselves and implement intransitive logic. DCs are able to couple unrelated chemical reactions to each edge. The output depends upon which node is used as input. DCs can also undergo selection to minimize the loss of thermodynamic entropy while maximizing the gain of information entropy. The intransitive logic underlying DCs enhances their programmability and impacts their evolution. The natural selection of DCs favors the persistence, adaptability, and self-awareness of living organisms and does not depend solely on changes to coding sequences. Rather, the process can be RNA-directed. I use flipons, nucleic acid sequences that change conformation under physiological conditions, as a simple example and then describe more complex DCs. Flipons are often encoded by repeats and greatly increase the Kolmogorov complexity of genomes by adopting alternative structures. Other DCs allow cells to regenerate, recalibrate, reset, repair, and rewrite themselves, going far beyond the capabilities of current computational devices. Unlike Turing machines, cells are not designed to halt but rather to regenerate.

## 1. Introduction

Contrary to widely held perceptions, what has been called junk DNA provides an evolutionary advantage by expanding the Kolmogorov complexity of genomes. Those repeat elements that adopt alternative DNA conformations under physiological conditions, called flipons, potentially enable new adaptations by altering the flow of genetic information into RNA. A subset of genomic repeats enables the assembly of novel cellular machines by encoding peptide patches that vary over time in length, composition, and chromosomal location. They promote the protein interactions essential for cells to regenerate, recalibrate, reset, repair, rewrite, and reproduce themselves into the next generation.

Normally, the analysis of how cells evolve focuses on the linear pathways (LPs) that connect substrates with products and the regulatory mechanisms involved. The evolution of systems can instead be viewed from a different perspective. This approach is based on directed cycles (DCs), in which all nodes are connected and where the path between adjacent nodes is directional. In a DC, the path taken between nodes depends on which node act as the input (Figure 1). There are a minimum of two directed paths between each pair of nodes that enable DCs to regenerate themselves. Each path can couple with different cellular processes to produce unique outputs. As I will discuss, the logic is intransitive, in contrast to the LP approach, where the logic is transitive. The design greatly increases the computations that a system can perform, a concept captured by the Kolmogorov complexity of a system, which is a fundamental measure of the information encoded by that system. To understand DCs in biological systems, it is also necessary to analyze them from a thermodynamic perspective and the way natural selection serves to optimize their energy efficiency. In this view, DCs are primary units of evolution that enhance the adaptability of cells through the computations they perform. These self-referential circuits act to contextually optimize responses. From a practical perspective, interventions targeting DCs will help in the design and engineering of new therapeutics and the development of bioprocesses that deploy specific chemistries.

This review is written from an information-theoretic perspective. It details the important role that DCs play in the evolution of living organisms. I start by introducing the logic underlying LPs and DCs. After providing examples, I discuss how DCs are implemented and powered in biological systems while incurring minimal losses to entropy. The strategies that cells use to ensure the regeneration of DCs are then discussed. Next, the intransitive logic enabled by DCs is explored, and the adaptability of these systems is examined from an evolutionary vantage point. The use of DCs by cells to monitor the self is then compared and contrasted with the properties of existing computing devices. Finally, practical applications of DCs in medicine and bioengineering are suggested that involve *in cellulo* evolution and the RNA-directed modulation of DC outputs.

## 2. Transitive versus Non-Transitive Logic

Transitive logic can be stated as follows: A > B and B > C imply that A > C where “>”indicates “greater than”. If the well-known rock, scissors, paper game used transitive logic, A would always beat B and C. You can guarantee a win by always playing A. With intransitive logic, the relationships are expressed as A > B > C > A with C > A. The intransitive relationship between A, B, and C is illustrated by the cycles drawn in Figure 1. These directed cycles (DCs) flow counterclockwise, as indicated by the direction of the arrowhead. When the rock, scissors, paper game is played with intransitive logic, C in fact beats A, despite A’s dominance over B and B’s dominance over C. The expectation of a win is one time in three if both players choose simultaneously. However, if the other player chooses first, you can always find an option that wins the game for you. There are some responses that will win in one situation and lose at other times. The most direct path between the nodes A, B, and C depends on which is selected first. Each edge of the DC has a different outcome associated with it.

So, how would intransitive systems work in biology? Before I answer the question, I will use flipons to provide an example of a directed cycle (DC). I will give a brief historical introduction to these ideas, drawing on the dissipative structures (dΣ) first described by Prigogine [1]. These systems remain in a stable state despite being far from the point of chemical equilibrium. Naturally, we need to answer the question, how is this energetically possible? If a system is running downhill, why does it not hit rock bottom? I will then ask the question of how DCs are used by cells for computation arise and then arrive at some surprising conclusions about how dΣ evolve. The hypercycles of Eigen are but one example [2].

## 3. Directed Cycles

DCs are those systems where one path leads to another and finally connects back to the starting point (Figure 1A). The net flow is in one direction only. In a DC, an entry can be at any point, and an exit can be at any other point. There is no beginning or end. The various inputs and outputs to the cycle are indicated by lines numbered 1 to 5. Regulatory inputs are indicated by broken lines, and they can be negative, positive, feedforward, or feedback. The molecules A, B, and C are components of the cycle, while X and Y are not. Of course, directed cycles are not perpetual motion machines. They are not like the Penrose impossible staircase, where you can effortlessly always go up or effortlessly always go down, depending on your choice, but not expend any energy as you start and finish each lap at the same position [3]. DCs are different. They go only in one direction, and they require an energy input to keep going. The energy expenditure is indicated by the Willard Gibbs equation:
(1)Δ*G =* Δ*H − T*Δ*S*

The term Δ*G* represents a change in free energy *G*, and Δ*H* represents the work necessary to complete the cycle, while the term called entropy (Δ*S*), a measure of the system disorder produced by the DC, represents the energy lost to the environment at the particular temperature (*T* in degrees Kelvin) studied.

### 3.1. Directed Cycles and Dissipative Structures

Where does the energy to power DCs come from? dΣ represent low-entropy states relative to their environment: they remain highly structured despite the disorganization around them. dΣ extract energy from the surroundings to maintain order. The energy is available to them because they exist far from chemical equilibrium. dΣ maintain a stable state despite widely fluctuating inputs. In many cases, dΣ can exist in more than one stable state. Surprisingly, a seemingly small situational slither is sometimes sufficient to set off a state switch.

Dissipative structures come in many forms. Flipons, which require energy to drive the flip from B-DNA to Z-DNA, are one example (Figure 1B) [4,5,6]. The energy needed depends on the DNA sequence and the base modifications present. The alternating d(CG) sequence with 5-methyl cytosine flips easily under physiological conditions, while other purine/pyrimidine repeats form Z-DNA less easily. The energy to initiate the transition can be generated by RNA polymerases. The polymerase acts by releasing chemical energy from the phosphate bonds broken as nucleotide triphosphates are incorporated into RNA. The polymerase also generates mechanical energy by stressing DNA as it unwinds the two strands of the double helix.

The flip from Z-DNA back to B-DNA follows a different path. The energy accumulated as Z-DNA can be released in a number of different ways to power completely different types of unrelated events. For example, the flip to B-DNA can fuel a change in the chromatin state to enhance or inhibit the subsequent transcription of a gene. Alternatively, topoisomerases can relax Z-DNA back to B-DNA to dissipate the energy and prevent the freezing of flipons in the Z-DNA conformation. The reduction in tension decreases the risk of strand breakage and DNA damage at the single-stranded junction between B- and Z-DNA. A different role is played by Z-DNA and Z-RNA in the regulation of immune responses, where helicases provide the energy to induce the transition from B- to Z-DNA that can activate programmed cell death pathways [7,8,9,10,11].

*Example of a complex dissipative structure*. More complex dissipative structures can form in completely different ways. Many involve quite complicated chemical pathways. One famous example is the Belousov–Zhabotinsky (Be-Zh) chemical reaction (Appendix A). Rather surprisingly, the color of a solution changes as the reaction progresses from red to blue to blue to red and so on as the solution is mixed either by constant stirring or by diffusion [12]. The color changes reflect the different oxidation states of iron and are driven by the release of free energy as the reaction proceeds (Appendix A). Each color state represents a distinct phase. At a critical concentration, there is a step-like switch from one phase to the other. Similar chemical gradients were independently proposed by Alan Turing to underlie the patterning of biological organisms, such as that seen with zebra stripes and in angel fish [13], and also by Hans Meinhardt and Alfred Gierer in 1972 [14].

*Entropy Produced by Dissipative Structures.* As dΣ reiterate, the paths traced vary from cycle to cycle and never overlap. With time, as the number of paths followed (*w*) becomes greater, it becomes harder to go back and retrace the exact history of the progression. As the entropy of the system increases, we can only follow the system forward in time, but we cannot retrace its past. We cannot reverse the timeline. The change in the entropy *S* over time *t* is described by Boltzmann’s formula:(2)$$\frac{\Delta S}{\Delta t}=k\sum_{t=1}^{t=n}{ln(w}_{t+1})-ln(w_{t})$$

This relationship can be captured in another way by using the Lyapunov function to measure how the paths diverge over time and is given by the value λ. The system centers on a point for values of λ less than zero and becomes chaotic as λ becomes more positive. When λ approaches zero, a DC forms. An example described by Robert May in 1976 [15] based on population growth is given in Appendix A.

In these examples, we have cycles that reiterate over the same time and trace paths that are similar, but not exactly the same. They consume energy and produce entropy, remaining stable over long periods of time. The equations describing these cycles produce order for certain values of the input parameters and do not contain variables that introduce randomness. The equations are fully specified and produce either single, ordered outcomes characteristic of dΣ as they reiterate, or chaotic outcomes otherwise (Appendix A).

### 3.2. Directed Cycles and Thermodynamic Entropy

Ilya Prigogine’s work concerned the energy flux through dissipative structures (Figure 1, Equation (1)). The DC will run if Δ*G* is negative; otherwise, an input of energy is required to drive the cycle. For work to be performed during the cycle (i.e., Δ*H* is positive), then Δ*S* must also be positive. In other words, the increase in the order of the system is exchanged for greater disorder of the environment. Overall, energy is lost to entropy. We have already seen that an increase in entropy over time is inevitable from cycling alone. The energy must be replenished from somewhere to maintain the cycle. Of course, a source of free energy *G* must be available to make up for losses if the cycle is to continue running. This is why dΣ only exist in states far from equilibrium, where free energy is available to power them. It is also in these regions that life is possible. Plainly put, attaining thermodynamic equilibrium by reaching a state where Δ*G = −T*Δ*S* is fatal to living organisms.

The DCs on which life depends are systems that ideally minimize the entropy loss and maximize the work performed. Intuitively, there is a parallel in the way that fluids pass through a tube. The output is maximal when the flow of the liquid is smooth, with all paths close and parallel to each other, i.e., when the flow is laminar. In contrast, turbulence disrupts efficient operation by negatively impacting the output. DCs, however, go beyond this analogy. There is ongoing optimization to minimize energy loss. Living systems optimize efficiency by improving the way that DCs are structured. They evolve DCs over time to improve their chances of survival. Those that fail do so soon fade into the past.

### 3.3. Directed Cycles and Self-Regeneration

To remain stable, DCs in living organisms must regenerate all of the components that constitute a node. They are prone to break, as losses of key constituents are unavoidable. They are also tasked to produce materials consumed by other processes. They must balance their outputs with the inputs they receive. DCs allow cells to avoid the infinite regress that Bob Rosen noted in 1959 [16], where to make a component, you require an enzyme, and to make that enzyme, you need another enzyme, and to make that enzyme, you need another enzyme, etc. DCs are quite flexible and solve the matching problem in a variety of ways. They can receive inputs and produce outputs of components from any part of the cycle. There are many opportunities to procure parts that they cannot replace themselves. In some cases, a downstream input will eventually regenerate a missing upstream input as the cycle reiterates. The input could also be sourced from the environment, from another cell, from the output of another DC, or from other reactions (Figure 2). Each design ensures that a DC remains in balance.

DCs can capture energy at multiple steps (Figure 3A). They can drive steps in the cycle that are thermodynamically unfavorable by ensuring that the products from the DCs are kept at low enough concentrations to pull the reaction forward (Figure 2, dΣ_f_). One example involves channeling a product through a membrane so that it is not available to drive the reverse reaction. The production of proton gradients across mitochondrial membranes is based on this strategy (Figure 3A). The gradients created can then push ATP production. This is the design formulated for the chemiosmotic theory by Peter Mitchell [17].

Of course, there are DCs that not only regenerate a component but also output that component, as well (Figure 3B). One example noted by Tibor Gánti is the glyoxylate cycle [18], in which malate uses the energy available from acetyl-CoA to both regenerate and output itself from the DC.
(3)malate + 2acetyl-S-CoA + 3H_2_O → 2malate + 2H-S-CoA + [6H]

This design enables the evolution of a different DC with malate as an input (Figure 2, dΣ_c_). Many of these strategies are based on autocatalytic chemistries, as described in a recent review that gives the history of these discoveries [19].

DCs can also incorporate feedback loops, including negative ones, as shown by the dotted lines in Figure 1. The refinement keeps the cycle in balance so that there is sufficient energy to regenerate itself at every possible turn.

### 3.4. Directed Cycles and Cellular Compartmentalization

The compartmentalization within the cell both enables the regeneration of DCs and increases the efficiency of energy capture. The isolation of DCs can be achieved using membranes, as exemplified by mitochondria, or by using phase separation based on condensate formation, as illustrated by spliceosomes and a variety of other nuclear bodies [20]. The compartments help localize components to ensure the efficient transfer of materials from one step to another in a DC. The arrangement also reduces competition from other DCs in the cell. The compartments can also provide a different chemical environment from that present in the rest of the cell, increasing the efficiency of a particular reaction. This set-up can also protect the cell from a product of the cycle that would otherwise be toxic to the cell.

Micellar compartments made of lipids have attracted much attention, as they self-assemble and permit different chemistries on either side of the membrane. The design allows for the specific transport of materials through the barrier. Membraneless condensates instead enable the assembly of proteins in layers to perform different functions [21,22,23]. For example, the nucleolus has three layers, the central, fibrillar, and granular shells, each performing a different function [24].

In other situations, amyloid-like fibers form by self-assembly, creating surfaces that promote specific outcomes. These include the MDA5 filaments that form on long double-stranded RNAs to initiate interferon responses [25], as well as the CARD (caspase recruitment domain) proteins that regulate inflammation and apoptosis [26]. In other cases, the membraneless condensates are not driven by protein domains but by a peptide repeat that forms an intrinsically disordered region (IDR). The IDRs lie outside the ordered domains of a protein. They promote and prevent interactions between proteins. They can be modified independently of protein function to capture the current cell status.

The number of possible ways in which IDRs can vary depends on their length and their composition. Those IDRs containing repeats can form a number of different covalent adducts. When the repeats are identical, the IDR adducts provide an analog readout of a particular pathway. When the repeats differ, they can measure the relevant activity of different pathways to provide a composite value. The design enables the rapid formation and dissolution of protein assemblies in response to ongoing changes in the cell state. Most importantly, IDRs are genetically encoded and subject to natural selection. Those specified by repeat DNA sequences, often referred to as junk DNA [27,28], are highly variable due to a high error rate during replication due to polymerase slippage, recombination, and frequent breakage-and-repair cycles. The repeats also undergo a rapid spread throughout the genome by a variety of recombination, repair, and transposition mechanisms [29]. Overall, they facilitate the assembly of various cellular machines into DCs and expand the functionality of the cellular wetware.

### 3.5. Directed Cycles and Informational Entropy

DCs can also be viewed in a more abstract fashion, with each edge of the graph equating to a computation. For example, relationships between nodes can be described using Boolean variables to represent inputs and outputs. This approach uses transitive logic to describe the relationships between each pair of nodes. A much larger array of possibilities arises when the intransitive nature of DCs is considered. The computational description enables an information-theoretic perspective of evolution that highlights the highly adaptive nature of DC. In contrast to minimizing the Δ*S* of the chemistry underlying dΣ by natural selection, evolution maximizes the informational entropy *I* by increasing the number of paths between DC nodes and by coupling different chemical processes to each path. Both elaborations produce a diversity of possible outcomes.

The simplest description of DCs starts with Boolean logic based on the assignment of “0” and “1” to whether an input or output is absent or present. This assumption is reasonable for enzymes that show a sharp dependence on input levels and respond in a step-like manner once a certain substrate concentration is exceeded. The directed cycles can then be viewed as a series of logic gates through which AND, OR, and NOT functions are implemented. In this case, the transitive relationships between input and output nodes can be used to construct a truth table. It is also possible to establish many different conditional relationships from dΣ. For example, in Figure 1, the output from an input at 4 can be 5 or 3, depending on whether an inhibitory signal at X or Y is present. X and Y break the DC at a particular place to produce this result. Depending on the decay of these signals, a DC regulated in this way can then provide a short-term memory of exposure to X or Y, resembling memory events that are observed in neural tissues (Figure 1, dΣ_d_).

What makes a system based on DCs different from a computer that uses only transitive logic? For a DC, the relationship between an input and an output is only a subset of the logical operations that the dΣ can perform. With a purely transitive design, the wiring of the input to the output is fixed. In contrast, the intransitive logic of a DC allows a node to assume many different roles. The node can be both an input to the DC, an input to the next step in the DC, an output from the preceding node in the DC, or an output from the DC. There are many possible ways to depict the relationships due to the self-referential nature of DCs: mapping of a component to itself (dΣ_1_) or to another component that is not x ((dΣ_2_), mapping of a component that is not x to x (dΣ_3_), and mapping of a component that is not x to itself (dΣ_2_). While these mappings are all true, it is possible to take pairwise combinations that are, on the surface, contradictory. The mappings raise the timeless existential question, “does x cause f(x) or does f(x) cause x?” Other contradictory logical schemes can depict the mapping of a DC component to itself or not to itself (Figure 2B). Gödel noted similar problems with self-referential statements formulated according to the rules of Peano arithmetic [30]. In both cases, the questions applied to DCs, or the statements identified by Gödel, remain undecidable. An input different from those currently available is required to resolve the problem.

The DC truth table depends on the roles assigned to each node at a particular time. For each DC, the wiring is fixed, not the order in which information is processed. The upstream node defines the path to the downstream node with a different route taken when the role played by each node is reversed. The design allows the DC to sense the availability of cellular resources and utilize the path most responsive. Outputs from the DC then vary in a way that minimizes the overall energy cost to the cell. DCs do whatever is necessary for a cell to survive in the most efficient way possible.

In summary, DCs are self-powering, self-restoring, and self-repairing. They model what is happening inside and outside the cell through the availability of inputs and the level of outputs. By monitoring how well DCs respond to these perturbations, a cell becomes self-aware and adjusts appropriately to avoid adverse outcomes. The intransitive nature of DCs enables cells to behave in ways that our currently manufactured computational devices cannot.

### 3.6. Directed Cycles and Computation

Of course, current computational devices are designed to be universal Turing machines and can, in principle, execute any program [31]. Further, it should also be possible to evolve programs that allow a Turing machine to complete a particular task with the fewest steps possible. Many different approaches to genetic algorithms have been tried to achieve this goal. Essentially, different programs are implemented to perform a particular task. A metric is then used to find the subset that performs best. The selected few are bred together to produce progeny programs that then undergo further selection. Code mutations and cross-overs between programs can be made at the binary level by flipping bits or by exchanging code snippets. There are limitations to this approach that are defined by the Kolmogorov complexity *K*(*x*) of the code, where:
(4)*K_p_*(*x*) = *min_p_*[*length*(*p*):*f*(*p*) = *x*]
and *p* is the shortest program run by a universal Turing machine with unlimited memory that halts after outputting *x* [32]. Of course, only a system with sufficient complexity can produce the outcome *x*. The length of *p* depends on the resources available to the system. With more complexity, substrings in the system arise that have very low complexity. Interestingly, these low-complexity strings can generate outputs that are highly variable. A seminal example is given by the game of life that was proposed by James Conway [33]. The game is coded using four simple rules. Depending on the starting position of the active cells, many different states can arise. They include stable or oscillatory structures and others that move across the board, some in a repetitive manner. Consequently, *K_p_*(*x*) is incomputable, as there is no way of guaranteeing the existence of a Turing machine capable of identifying the shortest *f*(*p*) *= x* for any particular program, given that it is not possible to know when you have found the solution that solves Equation (4).

Likewise, junk DNA contains low-complexity strings that can enable complex outcomes: flipons and peptide patches are just two examples. The ability of flipons to adopt different conformations has a great impact on the number of available genomic states, increasing by 2*^n^* with the number *n* of active flipons, even though the insertion of the same repeat elements at different locations does not greatly increase the total complexity of the genome sequence. The rapid increase in complexity due to flipons creates a digital genome capable of switching the analog output from the genome by changing the mix of transcripts produced. There is nothing in Equation (4) that captures this possibility. This uncertainly adds to the uncomputability of the genomic Kolmogorov complexity, as there is no way of guaranteeing the existence of a Turing machine capable of identifying the shortest *f*(*p*) *= x* given a large number of alternative flipon conformations.

One approximation of *K_p_*(*x*) is based on the compressibility of *x*. The compression estimate varies with the algorithms used and the substrings selected from *x*. Another method is based on the generalized topological entropy *H*(*ω*) of an infinite sequence *ω* calculated by partitioning X into overlapping sub-words [34,35], with *p_ω_*(*n*) being the number of different *n*-length sub-words that appear in *ω* and *n* being defined by 4*^n^* + *n* − 1 ≤ |*ω*| ≤ 4*^n^* + (*n* + 1), with the overlap of words allowed. However, the formulation does not capture the increase in entropy due to the altered flipon state. The informational entropy *I*(*ω*) is *H*(ω) adjusted for the flipon conformation and is given by
(5)$$I_{n_{\omega }}^{\left(k\right)}\left(\omega \right)=H_{n_{\omega }}^{\left(k\right)}\left(\omega \right)+F_{n_{\omega }}^{\left(k\right)}\left(\omega \right)$$
Hnω(k)ω=1k∑i=nω−k+1nωlog4⁡pωii , Fnω(k)ω=1k∑i=nω−k+1nωlog2(min⁡(j(i),1−j(i))i
with *k* representing the length of sub-words *k* < *n* such that |*ω*| = *n* + *k* − 1. The term *j*(*i*) is the probability that a sub-word is in an alternative DNA conformation, where 0 ≤* j*(*i*) ≤ 1, and varies by context. Another possible approach to capturing the coding capacity of a genome is based on calculating the Kullback–Leibler divergence from a reference genome [36,37,38,39,40]. There is, however, no adjustment for the flipon-mediated effects on coding due to changes in isoform usage, transcript editing, and RNA modification.

## 4. Evolvability of Directed Cycles through Junk Sequences

The analysis suggests that junk DNA extends the Kolmogorov complexity of programs that can be generated by the human genome through its effects on the flow of information from DNA to RNA. In humans, much of the junk arises from endogenous retroelements (EREs) that have contributed sequences to over 50% of the genome. The rewriting of genomic information by EREs occurs through the reverse transcription of the RNAs they transcribe into DNA. While initially dismissed as genomic fluff, it is now appreciated that EREs are essential regulatory components of genes. Sequences derived from Alu retroelements, of which there are over a million copies in the human genome, can change both the splicing and polyadenylation of nascent RNA (for recent reviews, see [41,42]). The EREs involved alter the transcript produced, depending on the position of their insertion, by implementing simple programming rules to change how RNA is processed. Rather than all being hard-wired into the genome, the outcomes are soft-wired and conditional on context. The splicing cascade that programs the sex of flies indicates the potential complexity of these events [43].

Flipons also offer the opportunity to modulate their conformations to alter downstream events. The single-stranded regions they form in the DNA duplex as they flip from one conformation to another expose binding sites that allow the sequence-specific docking of small RNAs, especially those derived from the same family of repeats as the flipon. The binding of these small RNAs to flipons allows the targeting of the cellular machinery to these genomic locations to edit and modify the transcripts produced. The proteins involved can be generic and bind in a structure-specific manner. They need not specifically recognize any particular nucleic acid sequence. The assembly of these complexes is directed only by the sequence-specificity of the RNA. This design has a number of evolutionary advantages. Importantly, the RNA sequence space available to target flipons in a sequence-specific manner is much larger than for developing sequence-specific proteins, where problems with folding and loss of function constrain the span of possible protein variations.

Changes in the RNA space are also not all or nothing, so there is no loss of any adaptations that proved successful in the past. The altered processing just increases the number of isoforms produced. In contrast, protein variants abandon the previous versions. Similarly, the spread of flipons through the genome creates many possible ways to alter the local DNA conformation to generate variant transcripts. The digital nature of flipons enables many different combinations. It is unlikely that the genomes of any two cells are set identically. As a consequence, the selection of cells at the tissue level can enable the responses that are most adaptable to local stressors (recently reviewed [44]). Small RNAs that are transmitted through germ cells also have the potential to bootstrap embryonic development by modulating the flipon conformation during early embryonic development [45]. These effects are likely modulated through the extraembryonic endoderm, which induces highly conserved programs within the rapidly dividing embryo and could possibly involve the reverse transcription of these small RNAs into the extraembryonic genome.

### 4.1. Evolvability of Directed Cycles through Peptide Patches

There are other ways to evolve a directed cycle through junk DNA. The peptide patches I discussed earlier as part of the cell’s wetware can act as Velcro to pull proteins together to create new assemblies (Figure 4A). The output from one of the sequestered proteins then potentially acts as an input to another. Eventually, a self-sustaining cycle arises through a set of protein interactions that positively reinforce each other’s output. This strategy assumes that proteins are more multifunctional than is currently presented in textbooks. In reality, the patched-together proteins often contain multiple different domains. Though many domains have well-studied functions, others remain uncharacterized. With the patchwork design just described, peptides with no enzymatic function can create new opportunities to unmask proteins with multiple personalities and are able to perform unexpectedly. Frequently, experimentalists find the newly discovered properties of a well-characterized protein surprising. They then write papers entitled “Hidden protein functions and what they may teach us” [46] and “Protein moonlighting: what is it, and why is it important?” [47].

The new cycles established by patching proteins together may initially depend on inputs from the milieux to bridge any missing links. The Krebs cycle that we depend upon to extract energy from sugars likely developed in such a way. At an early stage, the reactions depended on environmentally derived metals for catalysis. More efficient reactions arose when binding sites for metals were incorporated into genetically encoded proteins. Many of these strategies based on autocatalytic chemistries, along with the history of this field, have been reviewed [19]. Even today, some DCs still rely on environmentally derived factors to function. The dependency on these essential nutrients is so complete that, without them, certain DCs fail to regenerate. Humans, for example, do not synthesize vitamin C, even though other organisms solved this biochemical challenge long ago.

### 4.2. Evolvability of Directed Cycles through Hypercycles

The evolution of DCs can proceed through the organization of self-replicating molecules connected in a cyclic, autocatalytic manner, as originally proposed by Manfred Eigen [2] (Figure 4B). Due to the way they interact, the cycles are self-propagating, with each cycle forming a node coupled to a larger cycle (Figure 4). The interactions between different cycles allow them to amplify themselves, each other, and the hypercycle. The hypercycles further favor systems that store the information necessary to continuously regenerate themselves (Figure 4B). In the simplest form, the earliest steps in a pathway did all that was required to produce a particular output. Steps were added that closed the circuit, leading to the self-amplification of that particular cycle. The cycle underwent further elaboration by connecting to other cycles that further assured their mutual perpetuation (Figure 1, dΣ_c_). The creation of genetic systems to transmit this information to subsequent generations was a natural consequence of hypercycle evolution.

### 4.3. Evolvability of Directed Cycles through Genome Duplication

The rewriting of directed cycles in DNA during evolution can occur in many ways different from those that Eigen imagined. There may be more complex processes involved. On occasion, genes may undergo duplication in ways that Susumu Ohno demonstrated were important during evolution [48]. Fortuitous mutations affecting the level of gene expression, the processing of transcripts, and the non-templated modification of proteins then altered the character of each duplicated gene. At some point, changes to one paralog or the other provided a selective advantage, leading to the creation of new DC variants.

Occasionally, whole genomes undergo duplication. Many plants have a history of expanding their genomes in this manner and are consequently highly polyploid. As a result, they have multiple copies of each gene [49]. The process allows DCs to be reconstituted in different ways or with different combinations to generate new elaborations. The process of genome duplication has also been observed in yeast following a sudden and adverse change in the environment [50]. The high mutation rates that accompany this process drive additional genomic diversity and the elaboration of DCs that enable their regeneration and the survival of progeny in the new environment.

### 4.4. Evolvability of Directed Cycles through Endosymbiosis

Another way to acquire all of the components necessary to make a new DC is simply by obtaining all of them in one step from another organism. With bacteria, this means gaining an entire operon where all of the genes required for the regulation, expression, and scripting of a cycle are organized into one DNA segment. These outcomes are enabled by bacterial conjugation, the prokaryotic version of sex first observed by Joshua Lederberg and Edward Tatum [51]. To do the same in eukaryotes would require a genomic organization similar to the operons of bacteria and a truly giant virus to transmit the much larger eukaryotic genes that embed all of the required information. It is now possible, using a variety of technologies, to introduce into cells large genomic assemblies with all of the genes required. The most extreme transplant of genes so far performed is the transfer of entire normal mitochondria to replace the defective ones transmitted to an embryo from a parent. Of course, the only reason that eukaryotes have mitochondria in the first place is that at one point in time, the whole set of DCs that another free-living organism had successfully evolved was subsumed to generate energy with available substrates. The most recent proponent of this idea was Lynn Margulis, who also noted that chloroplasts are endosymbiont cyanobacteria [52]. Even today, osteoclasts can source replacement mitochondria from osteomorphs to remain functional [53].

### 4.5. Evolvability of Directed Cycles through Bioengineering

Experimental approaches aimed at modifying DCs depend on first identifying the minimal set of components required for a DC to regenerate itself. Such studies can be performed in vitro by purifying each element and reconstituting a DC from these parts. These approaches helped elucidate many of the DCs, such as the Krebs cycle, involved in cell metabolism. These studies can also be performed using genetic approaches to identify the different DC components. Over the years, bacteria and yeast have proven particularly powerful in establishing many of the factors that modulate DCs in single cells.

Collectively, these approaches identify the proteins essential for regenerating DCs. The methods also uncover redundancies and scaffolds that enhance the performance and robustness of DCs (Figure 1, dΣ_b_). Further, the results inform on which DC steps can be modulated therapeutically. Drugs to break DCs are part of the pharmacopeia positioned to kill cancer cells. The targeting approach yields valuable insights into the differences between normal and diseased cells. This work identifies multiple pathways between nodes in normal tissue and those that are no longer present in cancer cells. The vulnerability of tumors arises due to mutations that inactivate one or more of the redundant connections between nodes. The tumors are then susceptible to drugs that target the residual pathway. The drugs and mutations synergize to selectively kill the tumor while sparing normal cells.

Drugs that induce synthetic lethality in tumors are important in the clinic. In many cases, tumors are able to mutate and become resistant to most drugs that are used as single agents. The tumors then continue growing [54]. A drug cocktail that targets multiple DCs to induce synthetic lethality through different pathways is often needed to thwart the escape of cancer cells from eradication. The challenges to curing cancers despite the high-precision targeting of molecules underscore the overall resilience of DCs in cells. The intransitive programming based on DCs enhances their adaptability. Winning strategies just require the rewiring of the path between two nodes (Figure 1, dΣ_b_).

The therapeutic potential to alter DC function by programming flipons with small RNAs exists. The interventions can be used to prevent the expression of an essential DC component, to regulate its processing, or to recode the amino acids in key functional domains. There is also the possibility of rewiring connections in DCs to improve their design to engineer new functions. The nature of DCs allows us to drive their evolution in cells and to bulk manufacture their outputs by cell culture.

The patchwork approach to generating new DCs also offers opportunities. We do not know how far this strategy can be pushed to engineer new DCs. Experimentally, we could ask whether we can tag well-folded functional domains with interacting peptide patches to Velcro together new protein assemblies with defined properties. Can we then evolve a DC with a desired output (Figure 4A)? Or can we expose existing DCs to alternative chemistries to create completely new reaction schemes that have never before existed in nature? Already, DCs have been adapted to use synthetic chemicals in preference to their natural substrates. For example, Madeleine Bouzon and Philippe Marlière substituted 4-hydroxy-2-oxobutanoic acid for the amino acids serine and glycine as a carbon source for one particular metabolic pathway [55]. We have no idea what nature can do when put to the test.

An underexplored area is the use of repeat-derived RNAs to build scaffolds. As shown by the assembly of the spliceosome, many proteins exist that bind to simple sequence motifs exposed on single-stranded RNAs. In principle, these motifs could be used in a combinatorial fashion to create novel RNA scaffolds on which to assemble existing proteins in a cell into new assemblies and then select for a phenotype of interest. The targeting of the cellular machinery to triplex-forming flipons by noncoding RNAs through this mechanism has been previously reviewed [41].

## 5. Summary

The genetic encoding of DCs ensures the transmission of successful adaptations to future generations. The inherent programmability of DCs enables the survival of individuals over short time scales. Each DC can undergo optimization as an organism finds its niche. Conceptualizing the DC as a major unit in evolution focuses on the way in which dΣ enable the adaptability essential to an organism’s survival. DCs trade energy for information while minimizing dissipation and death from entropic extravagances. Despite the perpetually fluctuating environment, DCs ensure stability by resisting change. They promote the evolution of new DCs through the chemistries they enable.

DCs are self-referential in that each component regenerates itself (f(x) → x) (Figure 2B). Paradoxically, the junk in the genome makes such complexity possible. As Andrei Kolmogorov proved, it is not possible to program anything more complicated than the length of the longest sequence available to code with. While programmable like a Turing machine [31], the purpose of DCs is not to solve a problem and halt [56]. DCs do not terminate. Rather, DCs work best if they never stop. As dissipative structures, DCs offer the best way to survive in the midst of chaos, but they do not guarantee eternal life. DCs embrace intransitivity. They maximize informational entropy while avoiding thermodynamic equilibrium. DCs are not just the cycles of life, but they also embed the logic of life.

## Figures and Tables

**Figure 1 ijms-24-16482-f001:**
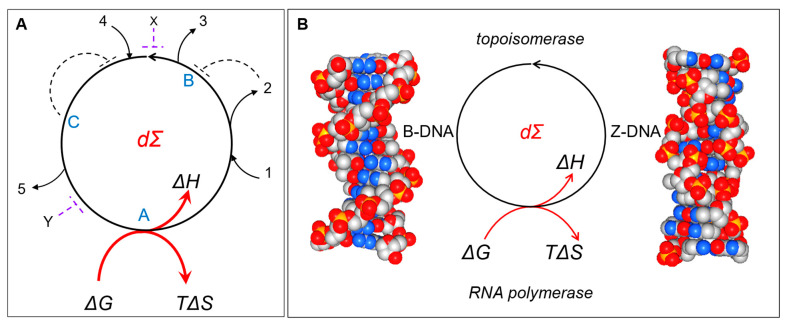
Directed cycle that implements intransitive logic. (**A**) It is possible to enter and leave the DC at multiple points. They each capture the relationship A > B > C > A. There is no beginning or end to the cycle. The letters A, B, and C could represent the rock, scissors, paper response and the numbers 1 to 5 refer to various environmental inputs and outputs. The cycle depends on the available energy (ΔG). The DC maximizes work (ΔH) by minimizing entropy loss (TΔS). The dotted lines represent a subset of possible paths that allow the negative regulation of the cycle through elements B and C or through points X and Y. In nature, these cycles are quite stable and can be described as a class of dissipative structures (dΣ). (**B**) Z-flipons are dissipative structures. They represent a DC for the transition between right-handed B-DNA and Z-DNA conformations. RNA polymerases can provide the energy to initiate the flip from B-DNA to Z-DNA. The energy cost depends on the DNA sequence and modifications to bases. The dissipation of energy by topoisomerases relaxes the Z-DNA to the B-DNA conformation.

**Figure 2 ijms-24-16482-f002:**
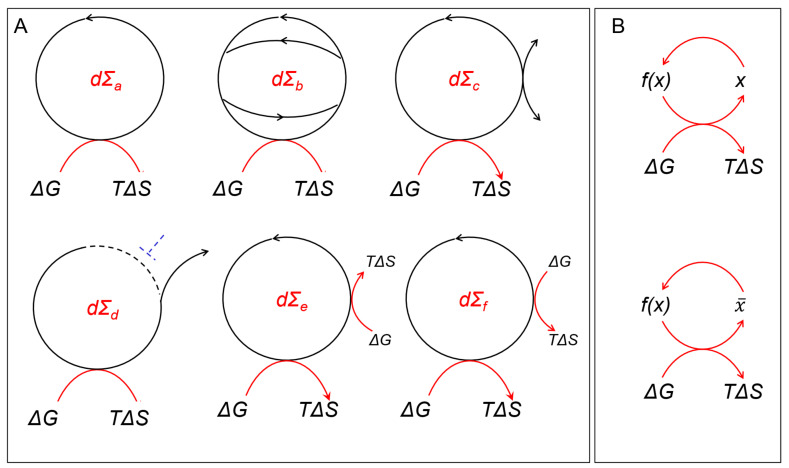
(**A**) Different ways to utilize dΣ are labeled with a subscript: a, reference; b, redundancy; c, connected to other DCs; d, inhibited so they produce a particular output; e, downhill to capture energy; f, uphill to enable energetically unfavorable reactions. (**B**) The different logical representations of DCs are all true, but pairwise, they can appear contradictory. With the information supplied, it can be proposed that *f*(*x*) maps to *x* or that *x* maps to *f*(*x*). Since the paths are not equivalent, the mapping may show that *f*(*x*) is causal for *x* or that it is not, but rather is causal for an intermediate step $\bar{x}$ that may or may not map to *x*. If the cycle looks messy, then you understand the point being made about how biological systems evolve.

**Figure 3 ijms-24-16482-f003:**
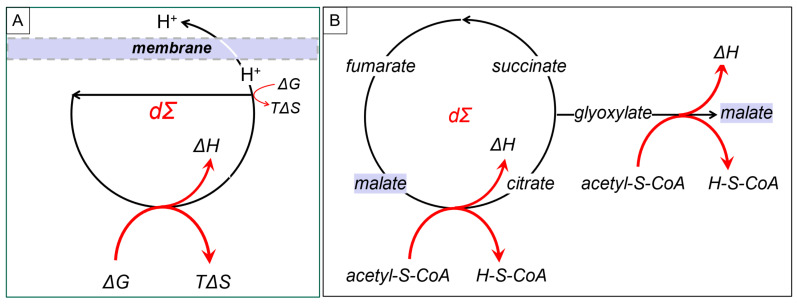
Powering unfavorable DC reactions (**A**). Transporting hydrogen ions across a membrane to pull an energetically unfavorable reaction (**B**). The glyoxylate cycle regenerates and outputs malate with acetyl-S-CoA, pulling both steps.

**Figure 4 ijms-24-16482-f004:**
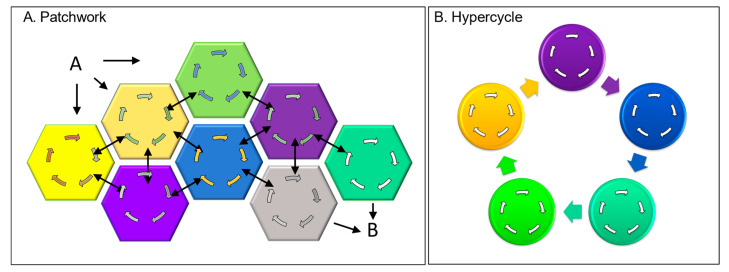
DCs as cellular building blocks. (**A**) Assembly of DCs into larger complexes through peptide patches, with the multiple connections between the input A and the output B increasing system robustness. (**B**) Interactions between directed cycles to produce hypercycles that are autocatalytic.

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
