# Peer review of "The Intransitive Logic of Directed Cycles and Flipons Enhances the Evolution of Molecular Computers by Augmenting the Kolmogorov Complexity of Genomes"

_ijms, 2023, doi:10.3390/ijms242216482_

Round 1

Reviewer 1 Report

Comments and Suggestions for Authors

I am having trouble following the main thrust of the paper. What makes the DC's so special? Is it their flexibility? Versatility? Robustness? There are many ways to increase Kolmogorov complexity in a network or in an organism.  What is so unique about DC's - or flipons - or hypercycles - or junk DNA?

The paper contains many details and examples, but I have not succeeded in seeing a coherent argument.

Comments on the Quality of English Language

There are several typos and grammatical errors. Here are some examples (there are more):

The ability of living organisms to survive (not organism)

Eigen (not Eisen)

In one step from another organism (not form)

by obtaining all of them in one step form another organism - should be to form another organism

Author Response

Thank-you for taking your time to read and review the manuscript.

I have made a number of changes to address your comments that I found to be quite helpful.

I have retitled the paper  so that it is more descriptive and I have rewritten the abstract and introduction to help the reader better orientate to the contents of the paper.

I have rewritten a number of sections to correct typos (thanks for pointing some out) and to help the presentation flow better.

I have also rearranged the order of text and created headings to make more explicit the importance of directed cycles, their intransitive logic, their robustness and how they contribute to evolution.

The sections in the manuscript address particular aspects of directed cycles from different disciplines. Overall, the manuscript spans thermodynamic. Information theoretic, biochemical, biophysical, genomic, and evo-devo disciplines. Each sections emphasizes the same overall theme but provides details relevant to a particular field. A reader can therefore sample what is most of interest to themselves.

Reviewer 2 Report

Comments and Suggestions for Authors

Reference Report

Title: Flipons and junk DNA contribute to the evolution of molecular computers by enhancing the Kolmogorov complexity of genomes

Manuscript number: ijms-2644033

Submitted to IJMS

By Herbert

The manuscript delves into how intransitive logic optimizes thermodynamic outcomes that support the persistence, adaptability, and self-awareness of living organisms. I have the following concerns regarding this topical review:

  1. It is recommended to utilize the formal IJMS template for presenting this manuscript. This will enhance the efficiency and expeditiousness of the review process.
  2. Abstract: While the abstract outlines the rationale behind this work, it does not sufficiently emphasize why this review is particularly timely.
  3. Introduction: The initial paragraph should be dedicated to the topic of the review's title. It's not advisable to delve into transitive logic in the first paragraph without establishing a clear connection to the central topic.
  4. Equation 1: The equation contains two labels that need clarification.
  5. Line 74: There should be substantive content following the question in order to provide a comprehensive answer.
  6. Line 40: In line 40, it mentions "dissipative systems (dΣ)," but in line 77, it reads "dissipative structures" (dΣ). Therefore, it is important to clarify whether the terminology refers to a system or a structure for consistency.
  7. Line 122: Please rephrase the following sentence: "An example described by Robert May in 1976 [15] based on population growth in given in Figure SF1F-H)" as it contains repetitive use of "in" with "given."
  8. I encourage the author to provide more insights into the future prospects of this topic, such as the potential utilization of machine or deep learning techniques for problem-solving.
  9. There should be two instances of "SF" in the manuscript, but the author has only provided one "scheme 1." Please ensure that both instances are included.
Comments on the Quality of English Language

No problem to read the manuscript but there are some careless mistakes.

Author Response

Thanks-you for going through the manuscript so carefully and for you thoughtful comments. My responses are belwo each point you make

  1. It is recommended to utilize the formal IJMS template for presenting this manuscript. This will enhance the efficiency and expeditiousness of the review process.

Completed

  1. Abstract: While the abstract outlines the rationale behind this work, it does not sufficiently emphasize why this review is particularly timely.

The review covers a lot of topics that are new to the discussion of evolution – including the role of flipons and the actual benefit of junk DNA in facilitating evolution. I am unaware of any discussion of evolution in terms of intransitive logic, nor any appreciation of directed cycles as a relevant unit of evolution. I have rewritten the abstract to highlight these points more clearly

  1. Introduction: The initial paragraph should be dedicated to the topic of the review's title. It's not advisable to delve into transitive logic in the first paragraph without establishing a clear connection to the central topic.

I wrote a new introduction

  1. Equation 1: The equation contains two labels that need clarification.

I annotated ΔG but the three other terms were already annotated

  1. Line 74: There should be substantive content following the question in order to provide a comprehensive answer.

I moved the question to after the heading as that is where the answer was given

  1. Line 40: In line 40, it mentions "dissipative systems (dΣ)," but in line 77, it reads "dissipative structures" (dΣ). Therefore, it is important to clarify whether the terminology refers to a system or a structure for consistency.

Thanks for catching this issue. Dissipative Structure is now used consistently throughout the manuscript

  1. Line 122: Please rephrase the following sentence: "An example described by Robert May in 1976 [15] based on population growth in given in Figure SF1F-H)" as it contains repetitive use of "in" with "given."

The first “in” was a typo and is now corrected to “is”

  1. I encourage the author to provide more insights into the future prospects of this topic, such as the potential utilization of machine or deep learning techniques for problem-solving.

The final section is now devoted to biotechnology and therapeutic applications

  1. There should be two instances of "SF" in the manuscript, but the author has only provided one "scheme 1." Please ensure that both instances are included.

Originally there were tow Supplementary Figures but now there is only one. I removed the reference to SF2.

Round 2

Reviewer 1 Report

Comments and Suggestions for Authors

I would like to see this paper published. However, I still do not totally understand the author's main argument. What makes DC's or Flipons so special, as compared to other network models such as catalytic networks, reflexively autocatalytic sets, hypercycles, etc.?

There is a basic assumption that the Kolmogorov complexity is an essential criterion. Perhaps, but this is not universally agreed. There have been many discussions on this and other criteria have been suggested. I would expect the paper to introduce the Kolmogorov complexity to the uninitiated and discuss its relevance.

There are a lot of ideas floated here, but the references are few. There have been many works discussing hypercycles since that 1971 reference. There have been so many works on chemical logic gates, catalytic networks, autocatalytic sets, etc. These should be referenced.

Comments on the Quality of English Language

There are still some typos. For example, "you require an enzyme", not "you require and enzyme". Also the style of the titles and subtitles should be consistent (periods? no periods?).

Author Response

I would like to see this paper published.

Agreed

However, I still do not totally understand the author's main argument. What makes DC's or Flipons so special, as compared to other network models such as catalytic networks, reflexively autocatalytic sets, hypercycles, etc.?

I rewrote the introduction to make it clear that the paper is going far beyond earlier descriptions of “catalytic networks, reflexively autocatalytic sets, hypercycles, etc.” by using the intransitive logic framework (which I think is easy to understand form the rock, paper, scissors intuition) and how that relates to the Kolmogorov complexity.

I changed the second paragraph to make it the first and rewrote the following introductory text. Hopefully the reviewer will find the intent of the paper clearer.

There is a basic assumption that the Kolmogorov complexity is an essential criterion.

I don’t understand the reviewer. the Kolmogorov complexity is not used as an essential criterion. The measure (or rather an estimate of the measure as explained in the text) is a fundamental way of describing the algorithmic complexity of a system.

Perhaps, but this is not universally agreed. There have been many discussions on this and other criteria have been suggested. I would expect the paper to introduce the Kolmogorov complexity to the uninitiated and discuss its relevance.

I added a further description in the introduction, but the concept is also described in simple text and by formula as such

“There are limitations to this approach that are defined by the Kolmogorov complexity K(x), of the code

where: Kp(x)= minp[length(p):f(p) = x]                      4

and p is the shortest program run by a universal Turing machine with unlimited memory that halts after outputting x”

There are a lot of ideas floated here, but the references are few. There have been many works discussing hypercycles since that 1971 reference. There have been so many works on chemical logic gates, catalytic networks, autocatalytic sets, etc. These should be referenced.

I think the reviewer will find that references 18 and 19 are pretty comprehensive reviews of the field.

  1. Gánti, T., The principles of life. Oxford University Press: Oxford ; New York, 2003; p 201 p.
  2. Bissette, A. J.; Fletcher, S. P., Mechanisms of autocatalysis. Angew Chem Int Ed Engl 2013, 52, (49), 12800-26.

If the reviewer feels there are other papers that should be referenced, it would be helpful for the reviewer to suggest them. The focus of the current review is not on the specific chemistry or ways people imagine how that arise, but rather on the thermodynamic and informational entropy involved and how that impacts the evolution of living systems. I am not sure that anyone has discussed junk DNA before in these terms before – especially not in terms of the binary encoding enabled by flipons and the effect on the Kolmogorov complexity of genomes. It is not my intent to deny someone credit for their work so I am happy to add appropriate references that would help the reader explore the literature further.

Reviewer 2 Report

Comments and Suggestions for Authors

I am satisfied with the modifications and corrections made by the author as per my comments.

Comments on the Quality of English Language

No problem to read this manuscript.

Author Response

Thank-you for your help with the manuscript!